# Temporal Evolution of Pressure Generated by a Nanosecond Laser Pulse Used for Assessment of Adhesive Strength of the Tungsten–Zirconium–Borides Coatings

**DOI:** 10.3390/ma14237111

**Published:** 2021-11-23

**Authors:** Joanna Radziejewska, Agata Kaczmarek, Tomasz Mościcki, Jacek Hoffman

**Affiliations:** 1Faculty of Mechanical and Industrial Engineering, Warsaw University of Technology, Narbutta 85, 02-524 Warsaw, Poland; jora@meil.pw.edu.pl; 2Institute of Fundamental Technological Research, Polish Academy of Sciences, Pawinskiego 5B, 02-106 Warsaw, Poland; akaczmar@ippt.pan.pl (A.K.); tmosc@ippt.pan.pl (T.M.)

**Keywords:** laser pulse, shock wave, MS coatings, ternary borides, laser adhesion test

## Abstract

The article presents theoretical and experimental study of shock waves induced by a nanosecond laser pulse. Generation of surface plasma pressure by ablation of the graphite absorption layer in water medium and shock wave formation were analyzed theoretically and experimentally. The amplitude and temporal variation of the shock wave pressure was determined basing on a proposed hydrodynamic model of nanosecond laser ablation and experimentally verified with use of a polyvinylidene fluoride (PVDF) piezoelectric-film sensor. The determined pressure wave was used for examination of adhesive strength of tungsten–zirconium–boride coatings on steel substrate. The magnetron sputtered (MS) W–Zr–B coatings show good adhesion to the steel substrate. The obtained experimental results prove the correctness of the proposed model as well as the suitability of the procedure for assessment of adhesive strength.

## 1. Introduction

The shock waves create many unique possibilities in materials testing. Generating pressure load as a result of applications of a high-energy laser pulse that caused metal surface evaporation was invented in the nineteen-sixties [1], and it was verified for metals with an uncovered surface in direct regime [2]. Such experiments required lasers with high pulse energy, which limited the application of this method in practice.

Application of inertial layer being transparent for the laser beam, confined regime, allows creating of the shock waves with high amplitudes, causing compression stress exceeding the yield point of metals [3,4]. The influence of different materials of inertial and absorption layers [5,6] on the profiles of generated pressures waves were studied extensively in the 1970s [7,8,9]. High stress waves induced by laser pulse give the possibility of changing the microstructure of materials [10]. This finding quickly led to extensive research onto the use of laser shock processing (LSP) as an alternative method for conventional shot peening and deep rolling [11].

High-pressure shock waves generated by laser pulse create many unique possibilities in testing of materials [12,13]. A broad range of pressures, speed, and deformation settings may be achieved as a result of changes in energy, shape, or duration of the laser pulse. On the basis of the laser shock waves, new diagnostic methods of dynamic behavior of material and layer [14], as well as adhesion of thin films could be developed [15]. Thin films are essential components of many microelectronic, optical, and micromechanical systems. During their manufacturing, a large amount of residual stresses is induced which influences their mechanical properties. In certain conditions, the residual stress may cause layer delamination from the substrate or its cracking. Several practical techniques of measuring the adhesion of thin layers are used. The most well-known are the scratch test as well as peel, pull, blister, and indentation tests. The Laser Spallation Technique was first introduced in the nineteen-seventies [16]. In this method, a layer is loaded with the stress wave created by short laser pulses. Adhesion of layers is determined on the basis of knowledge of a value of velocity of the layer during delamination. The accuracy of the method depends mainly on the precision of measuring of velocity of the back sample’s surface. Due to the very short time of the process, from several to tens of nanoseconds, highly advanced measurement techniques are required. The phenomena accompanying the delamination process have been widely studied in [17,18,19]. The technique was called LASAT (LAser Shock Adhesion Test). Soon, a combination of the laser shock waves generation and laser measuring was elaborated [20], which is still very useful in scanning analyzing of structures [21]. The LASAT tests were examined in many systems, including adhesion of different thin films, plasma layers, joined materials, or composites. Most of the works analyzed adhesion of thin films of about few micrometres thick [22,23,24,25,26]. The results have shown that LASAT should be assigned for evaluation of the adhesion of layers for the thin substrate. When crosswise dimension of the whole system is too large, pressure of the shock wave, during propagation in the sample, decreases due to a short-lasting pulse, so stress that causes delamination of layers becomes too weak. This limitation was solved using a laser of higher power in one pulse, at the level of several Joules [27,28]. In recent years, the interest in using laser pulses to test materials and layers continues.

This measurement methods were used for testing materials and layers, such as TiN [29], hydroxyapatite [30], thermal barrier coatings EB-PVD TBC with the use of shock wave propagation in two dimensions (LASAT 2D) [31], or carbon fibre reinforced composite CFRP [32]. Also, the surface shapes of the substrate and configuration of samples have been analyzed in detail [33,34].

Pulsed Laser Ablation in Liquid (PLAL), usually in water, supplies suitable conditions for high-pressure shock wave creation at relatively low laser fluences. For full absorption of the laser radiation the investigated metallic samples are usually coated with a thin layer of absorber. Additionally, such a layer isolates the investigated metal from ablation and other thermal effects associated with interaction of the laser beam. Hence, the investigation of underwater ablation of the graphite layer is important for underwater laser shock processing. Despite numerous uses, laser-induced underwater plasma physics is not fully understood. Especially results in the early phase of plume formation when pressure in plasma is the biggest are scarce. Low number of publications concerning plasma parameters is a result of diagnostic difficulties in case of dense, low-temperature plasma. Theoretical modeling is even more lacking. The first theoretical model of high-intensity shock waves by laser plasma in the water-confinement regime was presented by Berthe et al. [35,36]. A one-dimensional Lagrangian code, SHYLAC, was used to simulate the Al foil behavior under shock-wave loading. This code includes the hydrodynamic and elastoplastic response of the material and the water-confined regime laser interaction process description. Very good compatibility with the experiment was obtained by introducing the coefficient of efficiency of the interaction α. α is a parameter which defines the fraction of the plasma internal energy devoted to the pressure rise of the laser plasma and was appointed on the basis of experimental results [35,36]. The predictions of laser-induced shock pressures have been proposed and developed based on the confined ablation model also by Morales et al. [37]. The plasma dynamics was simulated by the one-dimensional radiation-magneto hydrodynamics code (HELIOS). By the HELIOS, the influence of the confining layer (medium and thickness) on plasma pressure in the case of aluminium target was studied. Hoffman et al. [38] presented a theoretical model of plasma formation during laser ablation of graphite in water. However, in this work the distribution of density and temperature in carbon plasma plume is presented but there is no information about distribution of plasma pressure on the surface of the target.

This article presents a study on shock waves formation and propagation in steel substrate with graphite absorption layer. The theoretical model of generation of initial surface pressure is proposed. The correctness of predicted plasma pressure in liquid environment was validated by shock wave pressure measurement by polyvinylidene fluoride (PVDF) piezoelectric-film sensor on the back surface of a tested steel plate. The obtained pressure values were then used to measure the adhesive strength of W–Zr–B layers deposited by the magnetron sputtering method. Novel W–Zr–B coatings were chosen due to their excellent mechanical properties i.e., very high hardness and relatively low Young’s modulus, which makes them competitive with commercial protective TiN coatings.

## 2. Materials and Methods

### 2.1. Materials and Coatings

Samples of the 15-mm diameter and thickness of 0.3, 0.5, 1.0, 1.5, or 2.0 mm were made from high speed steel HSS-SW7M/1.3343/HS6-5-2C (Pafana, Pabianice, Poland). Chemical composition of the material is shown in Table 1. This steel is used for tools that require high ductility, e.g., twist drills, thread cutting tools, relieved cutter, pull broach, reamers, and some cold work tools, e.g., punching punches.

Surfaces of the samples were ground and polished using diamond suspensions Before deposition process surface roughness was measured on confocal microscope VK-X100 (Keyence, 2800 Mechelen, Belgium) according to [39]. Values of roughness parameters were as follow: Ra = 0.024 µm, R_p_ = 0.07 µm, R_z_ = 0.133, and R_Sm_ = 141 µm.

The W_0.76_Zr_0.24_B_2.5_ target [40] was mounted on the water-cooled 1-inch magnetron sputtering cathode (TORUS Magnetron Sputtering Cathode, Kurt J. Lesker, Jefferson Hills, PA, USA). Deposition parameters were chosen on the basis of [41] and were as follows: initial and working pressure values were 2 × 10^−5^ mbar and 9 × 10^−3^ mbar, respectively, the gas flow of argon was 19 mL/min, power supplied to the magnetron cathode was 50 W. Each film was deposited for 120 min on high speed steel HSS samples heated up to 520 °C and positioned 40 mm in front of the target.

In order to ensure surface purity of the target and provide stability of sputtering conditions, the target was sputtered for 5 min prior to each deposition.

After the deposition process, the surface quality and thickness of the film were checked on a confocal microscope. The thickness estimated by the step method was 2.33 ± 0.14 µm. An example of the measurement result is shown in the Figure 1a. Roughness of the surface after film deposition slightly changed. Values of roughness parameters were as follow: R_a_ = 0.025 µm, R_p_ = 0.09 µm, R_z_ = 0.175, and R_Sm_ = 160 µm. Higher values of picks amplitude R_p_ and maximum roughness R_z_ were due to the presence of single particles on the surface visible in Figure 1b.

Nanoindentation analyses were performed at room temperature using NanoTest Vantage (Micro Materials Limited, Wrexham, UK) with Berkovich-shaped diamond indenter calibrated before each measurement with Diamond Area Function. Each experiment has been repeated 12 times in the line with a distance of 50 μm from each other. To minimize the effect of the substrate, the hardness and elastic modulus values were calculated based on the average data obtained at depths of about 100 nm (indentation load of 10 mN). It was done to ensure the maximum indentation depth of 1/10 of the coating thickness and thus minimize substrate effect and measure hardness in the load independent region. Values of hardness and Young’s modulus were as follows: H = 43.6 ± 0.7 GPa, E = 415.6 ± 6.5 GPa.

### 2.2. Shock Wave Pressure

Experiments were carried out with Nd:YAG pulse laser (981E, Quantel, Le Ulis Cedex, France) with a wavelength of 1064 nm, a pulse duration of 10 ns, and energy of 0.5 to 1 J. A spot diameter of 2.5 mm was used. The energy density that hit the sample surface was changing from 10 to 20 J/cm^2^, while the laser power density varied from 1 to 2 GW/cm^2^. The carbon layer was sprayed on irradiated surface of the sample to facilitate absorption. The average thickness, measured by confocal microscope, was about 5 µm.

Figure 2 shows a schematic of pressure measurement system. The set-up consists of five elements: a layer of water (1), graphite film (2), steel sample plate (3), polyvinylidene fluoride (PVDF) piezoelectric-film sensor (4), and (S_25CP Piezotech, Lyon, France) and Teflon disc (5).

Water serves as a transparent confining medium, whereas graphite is an absorber. Teflon disc placed under the sensor is an energy absorber. Small amounts of paraffin were used to improve the mechanical contact between the successive elements.

A PVDF sensor is placed on the back of the sample. However, registered pressures are lower than those occurring at the front surface of the sample. This phenomenon is caused by energy loses of the shock wave during propagation through the sample. In order to estimate pressure values of the shock wave at the front side of the sample, samples with different thicknesses were used. The estimation was made on the basis of an attenuation process [42,43].

Set-up characteristics and methods of calculations were presented in detail in previous studies [44,45,46]. The value of stress was calculated from registered signals from PVDF sensor. Results from PVDF sensor were compared with the measurements of velocity of back sample surface by VISAR (Velocity Interferometer System for Any Reflector). A qualitative compliance between the PVDF’s pressure and VISAR’s velocity rescaled to pressure was achieved [47].

The experimental set-up was modified to measure film adhesion to the substrate. In this case, there PVDF sensor was removed. Tested thin layer was on back side of steel plate and contact with atmosphere and the surface could deform freely.

### 2.3. Theoretical Model

The temporal variation of pressure that is induced on the surface of the sample and next generates a shock wave in it, can be theoretically modeled using the hydrodynamic model. The details of theoretical model were presented in [38]. During the first several microseconds after the beginning of the laser pulse the plasma is so dense, that it can be treated as continuum fluid and equations of gas dynamics can be applied for its description. The model which describes both the target heating, formation of the plasma and its expansion consists of equations of conservation of mass, momentum, energy, and the diffusion (species transport) equation. It is solved in axial symmetry with the use of the ANSYS-Fluent software package. The laser beam is normal to the surface and the focal spot on the target is 4.52 mm^2^. It is assumed that the plumes expand to water at ambient condition. The laser is a Nd:YAG laser operating at its first harmonic wavelength of 1064 nm with a pulse energy of 0.452–1.356 J and 10-ns pulse duration. The laser beam is focused on graphite layer with the laser fluence of 10–30 J·cm^−2^. For comparison for surface pressures the model of laser ablation in ambient argon is used, whose description was presented in [48].

## 3. Results and Discussion

### 3.1. Theoretical Surface Pressure

Figure 3 presents the temporal variation of maximal surface pressure on the sample surface during the first 50 ns as calculated in the model. The results for Ar ambient are also presented for comparison to show the differences in values of maximum pressure (Figure 4). In both cases, the shock wave creation is observed. For the target, which has been previously covered with a water layer, the plasma is confined and its expansion is delayed [38]. Therefore, the induced pressure is an order of magnitude greater (as 200 to 2000 MPa) and the pressure pulse duration 2–3-times longer than in the ambient gas (argon) [48] at the same power density. As it is shown in Figure 4, for laser fluence increasing from 6.5 to 15 J/cm^2^ the surface pressure increases almost linearly from 800 to 1900 MPa, respectively. It is due to growth of evaporation rate. After that, the absorption of a plasma plume is significant [38] what causes the surface pressure stabilization slightly below 2000 MPa. The second phenomena that can influence a growth of the plasma pressure is change of ablation regime. After reaching of critical parameters the phase explosion and fragmentation begin [49]. The maximum time duration of pressure above 1000 MPa is ~42 ns for 20 J/cm^2^ (Figure 3).

### 3.2. Experimental Measurement of Pressure

Unfortunately, it is not possible to measure pressure directly at the front surface of the sample. The sensor is placed on its back side. Hence, the comparison of pressure calculated in the model with measurements is not straightforward. It demands taking into account all energy losses during wave propagation. Firstly, there are partial reflections of wave at every boundary between the media: water, steel, and the sensor. Secondly, wave propagation in any medium is accompanied by energy losses. In our case a steel plate has the biggest thickness then wave attenuation in it must be included. The thicker the plate, the more it weakens the wave after it passes. Both issues are discussed in more detail in the following text.

Results of pressure measurement by a PVDF sensor registered for different thicknesses of the steel samples allowed estimation of pressure level on the front surface of the steel plate. The laser pulse and a waveform of pressure calculated from the PVDF sensor signal are presented in Figure 5. A series of subsequent pulses corresponding to shock wave reflection from the test plate back and front sides can be observed. Their amplitude gradually decreases as a result of attenuation in the plate.

The stress course for a system, with a steel plate that is 1-mm thick was used, is presented in Figure 5a. The maximum value of pressure at the first peak reaches 85 MPa. Time course of stress and amplitude are strongly modified by reflected stress waves. The velocity of sound inside steel plate estimated from experiment is about 5920 m/s.

Propagation of shock wave in real medium is accompanied by energy losses. Moreover, in a layered medium, the wave is partially reflected at boundaries. For a weak shock wave, the linear approximation holds, i.e., coefficients of reflection and transmission are functions of the ratio of acoustic impedances *A*, and given by Equation (1) [50,51]:(1)$$A=\frac{\rho_{2}c_{2}}{\rho_{1}c_{1}},R=\frac{A-1}{A+1},T=\frac{2A}{A+1};$$
where: *ρ*_2_*c*_2_, *ρ*_1_*c*_1_—acoustic impedances of contacting media (product of the density *ρ_i_* and the speed of sound *c_i_*); *T*—relative amplitude of wave transmitted from medium 1 to medium 2; *R*—relative amplitude of wave reflected from material boundaries.

Calculations of acoustic impedances were performed using the following data: steel–47 × 10^6^ kg/m^2^s, PVDF–3.8 × 10^6^ kg/m^2^s, or paraffin oil–2 × 10^6^ kg/m^2^s. The small impedance of the PVDF sensor in contact with the high impedance of the steel sample results in a five-fold decrease of pressure amplitude, i.e., pressure near the back side of the steel sample is five times the pressure measured by the sensor. In order to calculate pressure at the front of the sample, attenuation of the shock wave during propagation in steel has to be considered. Assuming an exponential damping formula after [42]:
(2)$$\sigma_{P}=P\cdot \exp\left(-n\cdot x\right),$$
where: *n*—wave attenuation coefficient; *x*—distance from front surface; *P*—initial pressure on the front surface at *x* = 0.

Estimation was made only for the first peak of the pressure registered by the sensor because amplitudes of subsequent peaks are distorted by reflected waves. In Figure 6, the dependence of pressure on the wave path in the steel (sample thickness) is presented. Points show values calculated from measurements, and the solid line is the fit of exponential formula from Equation (2) using the attenuation coefficient *n* = 0.8 mm^−1^ and the amplitude of the shock wave at the front surface P = 900 MPa.

Knowledge of the pressure value on the front side of steel plate allows for estimating pressure in confining medium. The pressure wave in water is much higher. Formula (1) shows that in water with small impedance (ρ_2_ = 1 g/cm^3^; c_2_ = 1500 m/s) that has a contact with steel with high impedance (ρ_1_ = 7.9 g/cm^3^; c_1_ = 5920 m/s), the amplitude of the shock wave is going to be two-times higher than inside the steel sample. The value of pressure wave in water near sample surface according to these calculations is equal about 1800 MPa for a 10-ns pulse with energy of 1 J. It should be noted that theoretical model of laser ablation was made for ideal conditions. Moreover, uncertainty in material properties and quality of contact between the sensor and the sample may affect the consistency of the results. However, the durations of first pressure peaks are comparable (~50 ns) that also confirms the compliance of the measurements with the theoretical model.

In Figure 3, the pressure registered at the back of 0.3-mm thick sample scaled according to the above procedure, alongside with theoretical pressure waveforms is presented. It was also necessary to shift it in time due to delays introduced by propagation in subsequent media. Unfortunately, such delays cannot be calculated with high accuracy due to lack of precise data concerning velocity of high amplitude wave. The experimental pressure waveform shows longer rise and fall times. It is due to dispersion in media in which the wave propagates, i.e., different propagation velocity for wave components.

### 3.3. Coating Adhesion Test

Tests were carried out for film W–Zr–B deposited on steel plates of thickness 0.3, 0.5, and 1 mm. Based on measurement results of the amplitude of pressure wave on the back side of the sample, the strength of adhesion was determined. The surface of samples after each test were observed and destruction of film was measured on the microscope. The tests show that in the case of the sample with 1-mm thickness there was no delamination. First, small deformation of the film after test was observed in the case of the 0.5-mm thickness plate. Measurable delamination was observed only for the sample with a thickness of 0.3 mm. In Figure 7, the surface of the film on the 0.3-mm sample after LASAT with 1 J is shown. The surface is flat and smooth, only a small local protuberance was observed probably in place of an imperfection of the film.

In the case of the 0.3-mm thick plate, delamination of the coatings can be visible (Figure 8). The maximum delamination height (determined in a reference to non-deformed film) was 1.6 to 2.5 um and the diameter was about 0.2 mm. No protuberance was observed when a laser pulse energy of 0.7 and 0.5 J was used. Adhesion can be estimated based on value of the maximum amplitude of compressive shock wave at the interface. Delamination is caused by tensile shock wave emerging due to reflection of compressive wave from the free surface. In case of the W–Zr–B film (ρ = 10.95 g/cm^3^, c = 6600 m/s [40,52]) on steel substrate the ratio of acoustic impedances is about 0.64. Relative amplitude of the wave transmitted from steel to the film is T = 0.78. In the case of 0.3-mm thick steel, the amplitude of pressure wave on steel/film boundary is about 700 MPa. The estimated amplitude of the tensile wave that caused coating delamination was about 320 MPa.

## 4. Conclusions

Theoretical and experimental studies of shock waves induced by a nanosecond laser pulse are presented. Generation of pressure induced by a nanosecond laser pulse ablation of absorption layer (graphite) in water medium and shock wave formation were analyzed theoretically and experimentally. The amplitude of the shock wave pressure was determined based on the proposed hydrodynamic model of nanosecond laser ablation and experimentally validated with the use of piezoelectric polymer PVDF sensors. The determined pressure wave was used for assessment of adhesive strength of magnetron sputtered, novel tungsten–zirconium–boride coatings on steel substrate. The main conclusions are as follows:The amplitude of the shock wave estimated based on the proposed hydrodynamic model was validated experimentally. Both methods showed good agreement. The maximal pressure in water near the sample surface can reach 2 GPa. Time duration of the first pressure peak is about 50 ns;Based on knowledge of shock wave pressure and acoustic impedance of tested materials, adhesive strength of coatings can be determined. The estimated amplitude of the tensile wave that caused W–Zr–B film delamination was about 320 MPa;The described above method is simple and does not require advanced equipment. The proposed method is less time-consuming than a traditional scratch test, which additionally needs greater surface to measure. However, the thickness of the sample is a limiting factor in the measuring range.

The proposed method is not intended to replace a traditional scratch test, but to augment the material testing toolbox. It helps to perform assessment of adhesion in dynamic conditions. We are currently planning more extensive tests of coating adhesion, carried out in parallel with two methods: scratch test and laser pulse.

## Figures and Tables

**Figure 1 materials-14-07111-f001:**
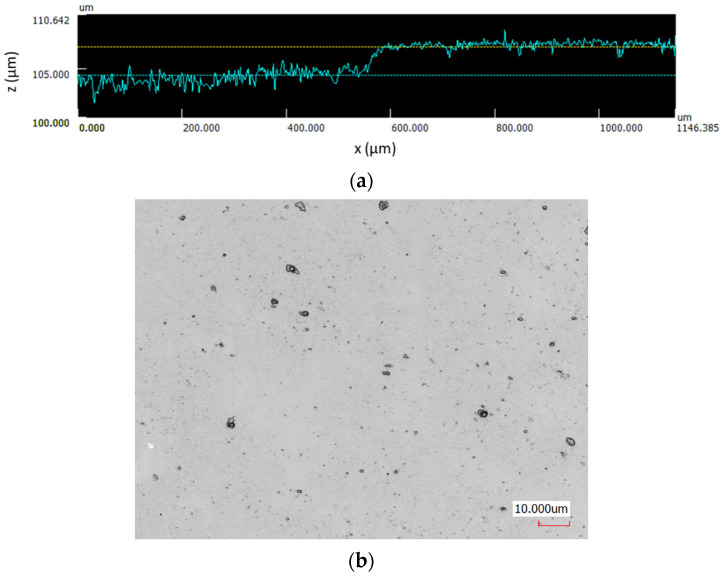
An example of the film thickness measurement (**a**) and surface quality (**b**) after the deposition process.

**Figure 2 materials-14-07111-f002:**
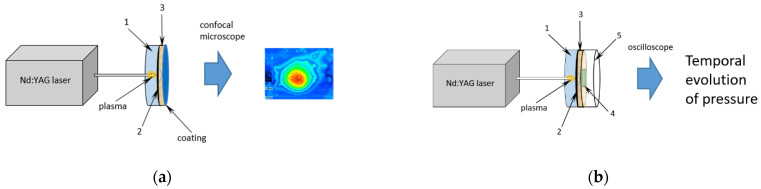
(**a**) Scheme of the experimental set-up for testing shock wave pressures induced by a laser pulse. 1—water, 2—graphite, 3—steel plate, (**b**) Modified set-up for measuring the film adhesion to the substrate by applying laser pulse. 1—water, 2—graphite, 3—steel plate, 4—PVDF sensor, 5—Teflon.

**Figure 3 materials-14-07111-f003:**
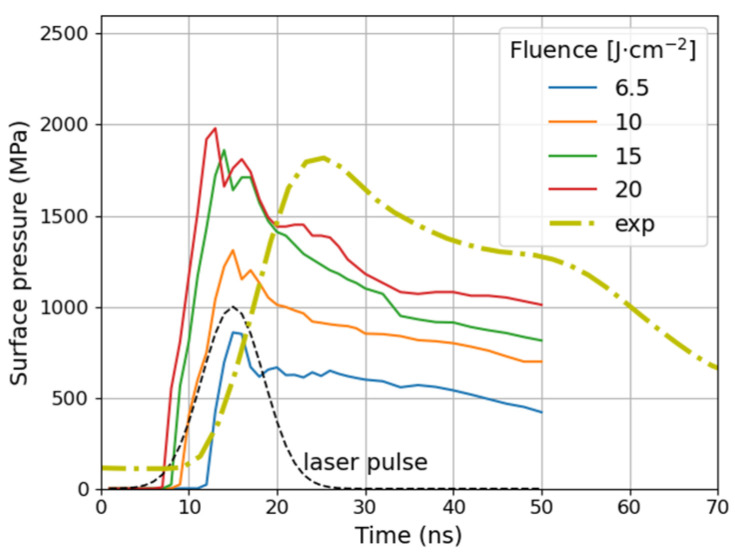
Evolution of surface pressure during 50 ns of plasma ablation process in confining medium (water) calculated in the model for fluences 6.5, 10, 15, and 20 J·cm^−2^. Experimental surface pressure labelled **exp** is also shown for a comparison. It was calculated from measurements as described in details in Section 3.2.

**Figure 4 materials-14-07111-f004:**
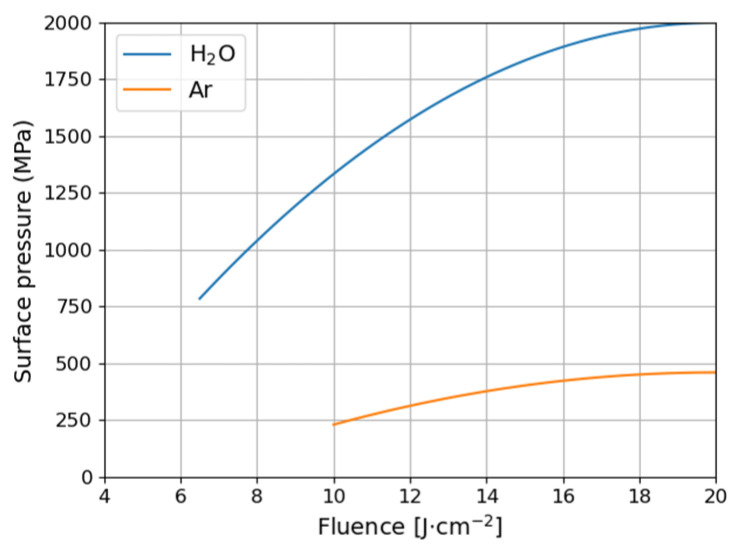
Maximum surface pressure as a function of laser fluence for water calculated in the model. Comparison for water and ambient gas Argon [48].

**Figure 5 materials-14-07111-f005:**
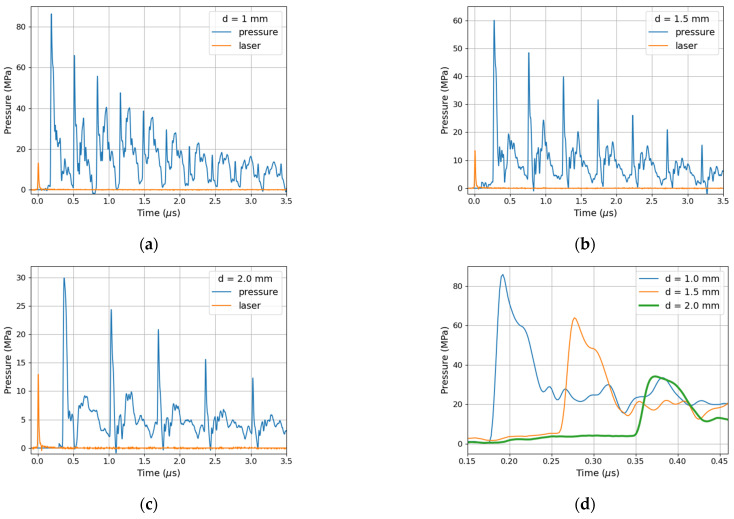
Temporal evolution of pressure registered by PVDF sensor at the back side of the sample. Sample thickness d = 1 (**a**), d = 1.5 (**b**), and d = 2 mm (**c**). Zoom of the first pressure peak for d = 1, 1.5, and 2 mm (**d**). Laser pulse energy of 1 J, pulse duration 10 ns.

**Figure 6 materials-14-07111-f006:**
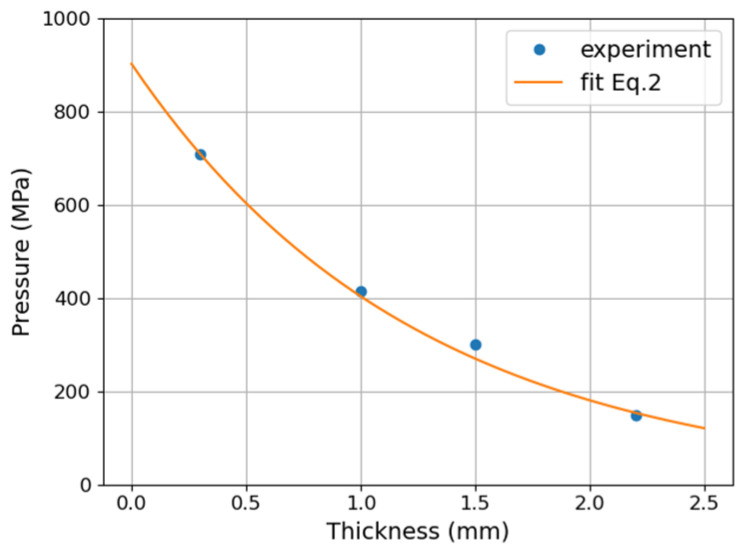
Pressure as a function of wave path in steel calculated from measurements made by PVDF sensors. Points—experimental values, Line—fitted function from Equation (2).

**Figure 7 materials-14-07111-f007:**
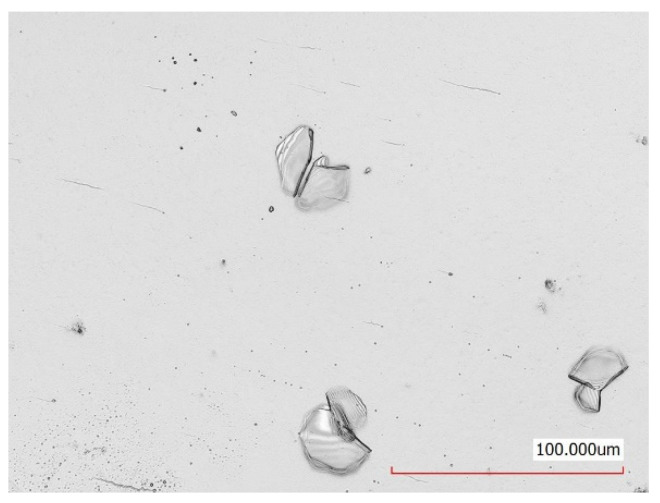
Surface of the 0.3-mm sample. W–Zr–B film after LASAT with laser pulse energy of 1 J and duration of 10 ns. Delaminated fragments of the W–Zr–B film are clearly visible.

**Figure 8 materials-14-07111-f008:**
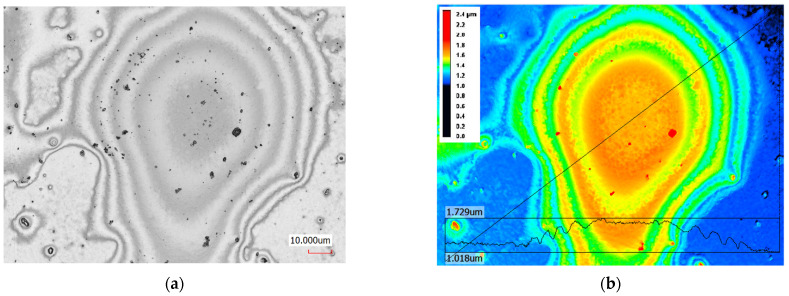
Surface and map with profile of the protuberance W–Zr–B film after delamination from steel substrate 0.3-mm thick with a laser pulse energy of 1 J and duration of 10 ns. (**a**) Surface image after layer delamination; (**b**) contour map and profile of the protuberance; and (**c**)contour map and profile of protuberance in the next sample delaminated at the same condition.

**Table 1 materials-14-07111-t001:** Chemical composition of HSS-SW7M/1.3343/HS6-5-2C.

C	Si	Mn	P	S	Cr	Mo	W	V	Co	Ni
0.86–0.94	Max. 0.45	Max. 0.40	Max. 0.03	Max. 0.03	3.80–4.50	4.70–5.20	5.90–6.70	1.70–2.10	-	-

## Data Availability

The data presented in this study are available on request from the corresponding author.

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
