# Peer review of "Temporal Evolution of Pressure Generated by a Nanosecond Laser Pulse Used for Assessment of Adhesive Strength of the Tungsten–Zirconium–Borides Coatings"

_materials, 2021, doi:10.3390/ma14237111_

Round 1
Reviewer 1 Report
The authors presented a new technique to measure the adhesive strength of a coating by a shock wave ns pulsed laser in water medium. This work focuses on the experimental side, but the theoretical modeling is presented elsewhere. The work is good, but lacks some explanations. Following points need to be considered before publication:
- Line 127: Please provide minimum deposition parameters rather than citing a reference. What is the film hardness?
- Figure 1a: Include the names of X and Y axis of the graphical image.
- Figure 5: Can the authors provide an explanation behind the pressure decrease with an increase in sample thickness?
- The pressure units are different in Figures 3, 4 and 5, 6. Please use consistent units of Pa/MPa throughout the manuscript.
- Line 271: Authors have mentioned the value of pressure wave in water is ~1.8 GPa for 10ns pulse and 1 J energy according to the theoretical model. What is the experimental value for the same? Figures 3 and 4 do not show any experimental value related to this to validate/ understand the closeness of the model to the experiment.
- Why only the 0.3 mm, 0.5 mm and 1 mm plate thicknesses are chosen for coating deposition? Why not the higher thicknesses like 1.5 and 2mm?
- Line 288: Not clear what authors meant to say about maximum height. Are they talking about diameter of the delaminated area?
- Which plate thickness is shown in Fig. 7? If this is 0.5 mm, then provide the image for the 1 mm thickness plate as well.
- Include the color bar in Fig 8. What are these three images showing? Include number and title of the images.
- Can authors shade some light on the adhesion of coatings deposited on 1.5 and 2 mm thicknesses? Do all thicknesses over 1 mm show no delamination? If yes, then why?
- What is the advantage of using this model over traditional scratch test expect the small area requirement? This method seems to have a limitation depending on the plate thickness.
Reviewer 2 Report
The authors have studied the possibility of using the technique of shock wave generation during laser ablation in water medium to assess the adhesion of coating to a substrate. To confirm the experimental results, they carried out theoretical calculations according to the mathematical model which they have described earlier. The presented results are of interest to specialists in the field of laser processing and coating production and may be published. However, several issues require clarification.
- The relevance of research in the field of laser irradiation under a transparent layer (water) is due to several reasons, in particular, the fact that the high-pressure region arising during such irradiation of materials provides an increase in the efficiency of surface laser alloying of materials (for example, Fominski V. Yu. et al. Surface alloying of metals by nanosecond laser pulse under transparent overlays. J. Appl. Phys. 93, 5989 (2003); doi: 10.1063 / 1.1568149).
- Figure 2 should contain two parts: a) Diagram of the experimental setup for testing shock wave pressures induced by a laser pulse and b) Modified diagram for measuring the film adhesion to the substrate by applying laser pulse.
- The information on how the carbon layer was applied and how thick it is should be added.
- It is desirable to prove that the profile shown in fig. 8 is due to delamination of the coating rather than deformation of the very thin steel substrate after pulse laser irradiation.
- It is necessary to give a description (for example as (a), (b) and (c)) to all images of Figure 8. Where was the bottom map/profile measured?
- Indicate the thickness of the steel substrate in the caption to Figure 7.
- The authors note several advantages of the laser technique for measuring the adhesion of coatings. It is desirable to show/discuss how the results of laser testing correlate with the results of traditional scratch test for the coatings obtained by the authors.
Round 2
Reviewer 1 Report
The comments have been addressed, except following:
- Can the authors include the experimental values as mentioned in comment no 5? The 1800 Mpa value is calculated from the model, not from the experiment. The authors have provided the explanation for Figure 3 but missed the same for figure 4.
- Can the authors explain comment no 10, that why there is no delamination for thicknesses over 1 mm?
Also change the sentence in line 355-356 in passive form, as it has been directly copied from review response.
